# Respiratory virus surveillance in hospitalized children less than two-years of age in Kenema, Sierra Leone during the COVID-19 pandemic (October 2020- October 2021)

**Robert J. Samuels**[1☯*], **Ibrahim Sumah**[1‡], **Foday Alhasan**[1‡], **Rendie McHenry**[2‡], **Laura Short**[2‡], **James D. Chappell**[2‡], **Zaid Haddadin**[2☯], **Natasha B. Halasa**[2☯], **Inaê D. Valério**[3‡], **Gustavo Amorim**[4‡], **Donald S. Grant**[1,5‡], **John S. Schieffelin**[6‡], **Troy D. Moon**[2,3,6,7☯]

1 Kenema Government Hospital, Ministry of Health and Sanitation, Kenema, Sierra Leone, 2 Department of Pediatrics, Division of Pediatric Infectious Diseases, Vanderbilt University Medical Center, Nashville, Tennessee, United States of America, 3 Vanderbilt Institute for Global Health, Vanderbilt University Medical Center, Nashville, Tennessee, United States of America, 4 Department of Biostatistics, Vanderbilt University Medical Center, Nashville, Tennessee, United States of America, 5 College of Medicine and Allied Health Sciences, University of Sierra Leone, Freetown, Sierra Leone, 6 Department of Pediatrics, Division of Pediatric Infectious Diseases, Tulane University, New Orleans, Louisiana, United States of America, 7 Department of Tropical Medicine, Tulane University School of Public Health and Tropical Medicine, New Orleans, Louisiana, United States of America

☯ These authors contributed equally to this work.
‡ IS, FA, RM, LS, JDC, IDV, GA, DSG and JSS also contributed equally to this work.
* robjsam190@yahoo.co.uk

**Data Availability Statement:** A deidentified data set has been uploaded to OSF at DOI:osf.io/2peh6

## Abstract

Globally, viral pathogens are the leading cause of acute respiratory infection in children under-five years. We aim to describe the epidemiology of viral respiratory pathogens in hospitalized children under-two years of age in Eastern Province of Sierra Leone, during the second year of the SARS-CoV-2 pandemic. We conducted a prospective study of children hospitalized with respiratory symptoms between October 2020 and October 2021. We collected demographic and clinical characteristics and calculated each participant´s respiratory symptom severity. Nose and throat swabs were collected at enrollment. Total nucleic acid was purified and tested for multiple respiratory viruses. Statistical analysis was performed using R version 4.2.0 software. 502 children less than two-years of age were enrolled. 376 (74.9%) had at least one respiratory virus detected. The most common viruses isolated were HRV/EV (28.2%), RSV (19.5%) and PIV (13.1%). Influenza and SARS-CoV-2 were identified in only 9.2% and 3.9% of children, respectively. Viral co-detection was common. Human metapneumovirus and RSV had more than two-fold higher odds of requiring O2 therapy while hospitalized. Viral pathogen prevalence was high (74.9%) in our study population. Despite this, 100% of children received antibiotics, underscoring a need to expand laboratory diagnostic capacity and to revisit clinical guidelines implementation in these children. Continuous surveillance and serologic studies among more diverse age groups, with greater geographic breadth, are needed in Sierra Leone to better characterize the long-

[https://osf.io/2peh6/files/osfstorage/
6493215b67aff805b3ee0303].

**Funding:** RJS, DSG, JSS, TDM U2RTW011248
Fogarty International Center of the National
Institutes of Health www.fic.nih.gov The funders
had no role in study design, data collection, and
analysis, decision to publish, or preparation of the
manuscript.

**Competing interests:** The authors have declared
that no competing interests exist.

term impact of COVID-19 on respiratory virus prevalence and to better characterize the seasonality of respiratory viruses in Sierra Leone.

## Introduction

A majority of school absenteeism, hospitalizations, and deaths in children under-five years of age in Africa is attributable to acute respiratory infections (ARI) [1, 2]. To date, much of the focus for ARI in developing countries has been on identifying bacterial causes such as *Streptococcus pneumoniae* and *Haemophilus influenzae* type b *(Hib)* in this age group [1, 3–7]. However, many low-resource settings lack the basic diagnostic capacity for identification of such bacterial and/or viral respiratory infections. As a result, a certain proportion of children admitted to hospital with presumed bacterial infection will fail antibiotic treatment, leaving clinicians frustrated and questioning their treatment decisions. Moreover, widespread antibiotic use in this population is likely associated with increased risk for the promotion of antimicrobial resistance [5, 8].

Globally, viral pathogens are now the leading cause of ARI, especially after the introduction of *Hib* and pneumococcal conjugate vaccines. Further, recent studies from several African countries have also identified viral pathogens as key contributors of ARI in children under five-years of age [7, 9, 10]. As the number of immunization programs for the common bacterial causes of ARI increases across sub-Saharan Africa, the proportion of ARI caused by respiratory viral pathogens has increased [1]. In these cases, viral ARI may cause symptoms alone or be associated with a co-infection with other viruses or bacteria, leading to severe pneumonia [1, 9]. The COVID-19 pandemic has been associated with changing trends in the prevalence of other respiratory viruses, possibly as a result of public health measures aimed at limiting circulation of SARS-CoV-2. However, this phenomenon is not yet fully understood and warrants continued vigilance as COVID-19 mitigation strategies are relaxed around the world [11–16].

Sierra Leone, located in West Africa has an estimated population of 8.6 million people as of 2022 and was ranked 182 out of 189 countries on the United Nations Development Program (UNDP) Human Development Index (HDI) for the same year [17, 18]. Health infrastructure was severely affected by civil war (1991–2002), and then again by the West African Ebola Outbreak (2013–2015) [19, 20]. Since 2006, Sierra Leone has established national-level influenza surveillance and began contributing to the World Health Organization (WHO) Strengthening Influenza Sentinel Surveillance in Africa (SISA) network in 2011 [21, 22]. Despite this, there remains a paucity of information in Sierra Leone related to the prevalence, temporality, and epidemiology of the most common and emerging human respiratory viruses.

The major aim of this study is to describe the prevalence and temporality of viral respiratory pathogens in hospitalized children under-two years of age, at one rural hospital in Eastern Province of Sierra Leone, during the second year of the SARS-CoV-2 pandemic and to characterize the social, environmental, and medical risk factors associated with severe disease.

## Materials and methods

### Study design and population

We conducted a prospective surveillance study of children admitted to the Kenema Government Hospital (KGH) pediatric ward between October 2020 and October 2021. Children were eligible for enrollment if they were <24 months of age, had been admitted to the hospital within 48-hours prior to enrollment, with or without fever of $\geq 38.0°C$, and presented with at

least one of the following respiratory symptoms: cough (<14 days duration); difficulty breathing (retractions, tachypnea for age: >50 breaths per minute for those 0–12 months of age and >40 breaths per minute for those 13–23 months of age), and/or nasal flaring. Patients who did not meet the above criteria were excluded from participation. Participants were only enrolled in the study once.

## Study setting

This study was conducted at KGH, in Kenema District of Eastern Province, Sierra Leone. KGH has a catchment area endemic for illnesses such as malaria and Lassa fever (LF) and is a hospital that was at the epicenter of Sierra Leone´s efforts during the 2014–2016 West African Ebola outbreak [19, 23, 24]. The pediatric ward of KGH provides free health care for children under five years of age and admits approximately 400 children per month. The ward uses the Emergency Triage, Assessment and Treatment plus (ETAT+) protocol for the management of patients. For example, children with oxygen saturation less than 90% are initiated on supplemental oxygen therapy. The ward has a unit that serves both as an emergency room and an intensive care unit. Admission to this unit is for patients with emergency and some priority signs as per the ETAT+ protocol [25]. Malaria, pneumonia, malnutrition, and gastrointestinal illness are the four main reported causes of hospitalization in the under-five age group. KGH has radiograph capacity on site but lacks both blood culture capacity and the ability to diagnose respiratory viral pathogens. Routine laboratory analysis is limited to simple hematology and chemistries; rapid tests for malaria, HIV, typhoid, and syphilis; microscopy for malaria and stool samples; as well as point of care technologies such as GeneXpert MTB/RIF (Cepheid: Sunnyvale, CA) for tuberculosis (TB).

## Recruitment

Upon admission, children with acute respiratory symptoms were pre-screened for potential eligibility by emergency care nurses and the hospital admission team. Once preliminary eligibility was established, parents or legal guardians were approached by trained study personnel for enrollment until a daily quota was met (approximately three new patients enrolled per day). Enrollment took place Monday-Friday during daytime work hours (6am to 6pm).

## Data collection

Data were collected by study nurses utilizing a paper-based study instrument and then uploaded into a password-protected, tablet-based, online Research Electronic Data Capture (REDCap) database (www.project-redcap.org). Parents or legal guardians were asked a series of questions using a standardized case report form (CFR). The CFR covered seven broad areas: Introduction and screening, demographics and social history, current illness, diet/nutrition, physical examination, and laboratory results. At admission, participants´ respiratory symptom severity was recorded using an adapted version of the bronchiolitis severity score published by Tal et al, with patients subsequently categorized as having mild, moderate or severe symptoms [26]. Chart reviews were done during admission and upon discharge, referral, or death. This method allowed for the recording of demographic information, medical and medication history, and information on clinical course while hospitalized. Data quality control was conducted by study investigators who reviewed all completed paper-based study instruments and confirmed the accuracy of data entered into the electronic database.

## Laboratory procedures

Nose and throat swabs were collected by the study team at enrollment and combined immediately in viral transport medium. Samples were labeled with the participant's identification number to ensure anonymity and stored at -80˚C (KGH) until shipment on dry ice (Nov. 2021) to Vanderbilt University Medical Centre (VUMC) for testing.

Total nucleic acid (TNA) was purified from aliquoted specimens using the QIAamp 96 Virus QIAcube HT Kit and QIAcube HT automated extraction system (QIAGEN, Germantown, MD) and maintained at -80˚C until analysis. TNA extracts were tested for the following pathogen targets using the *NxTAG Respiratory Pathogen Panel + SARS-CoV-2* (Luminex, Austin, TX): adenovirus (AdV); human endemic coronaviruses (hCoV) 229E, HKU1, NL63, OC43; SARS-CoV-2; human bocavirus (hBoV); human metapneumovirus (hMPV); influenza A and subtypes 2009 H1N1, H1, and H3; influenza B and influenza C; parainfluenza (PIV) 1–4; respiratory syncytial virus (RSV) A and B; human rhinovirus/enterovirus (HRV/EV); *Chlamydophila pneumoniae*; and *Mycoplasma pneumoniae*. Reverse-transcription quantitative PCR (RT-qPCR) on the QuantStudio 3 and QuantStudio 6 Pro Real Time PCR Systems was employed to test extracts for the following targets using *SuperScript III One-Step RT-PCR System* with current versions of assays developed at the US Centers for Disease Control and Prevention (*CDC Human Influenza Virus Real-Time RT-PCR Diagnostic Panel Influenza A/B Typing Kit*, *Influenza A Subtyping Kit*, and *Influenza B Lineage Genotyping Kit*; and *Influenza C Assay*): influenza A and subtypes, influenza B and lineage genotypes, influenza C, and human RNase P (endogenous target for specimen adequacy).

## Statistical analysis

Descriptive statistics were presented as frequency (percentage) or median and interquartile range [IQR] where appropriate. Categorical variables were compared using Pearson $\chi^2$ tests. Continuous variables were compared using Mann-Whitney U test. Multivariable logistic regression was used to find risk factors associated with higher odds of being referred to the intensive care unit (ICU) or requiring oxygen (O2) during hospitalization. Variables included in the final model were pre-specified and selected based on literature review and clinical knowledge. In order to avoid subjective discretization of a continuous variable, we analyzed severity score as an ordinal, continuous variable, and used a multivariable ordinal regression model with a logit link to regress it on the same covariates used in both logistic regressions mentioned above [27]. Data discretization refers to a process of converting large amounts of data values into smaller ones, so that their subsequent evaluation and management becomes easier. Continuous variables, such as age and birthweight, were modelled using restricted cubic splines, with three knots equally spaced to alleviate linearity assumptions. Interpretation of these non-linear effects on the outcomes used, contrasts between pre-specified age and BMI, as suggested by Shepherd and Rebeiro (2017) [28]. A propensity score approach was used to estimate the effect of each type of virus on all three outcomes (being referred to ICU, requiring O2, and severity score). More specifically, we defined a binary variable to identify patients that were diagnosed with a particular virus of interest. For instance, while focusing on AdV, patients that were diagnosed with AdV would be label as "1" while patients that did not have AdV detected in their samples received a value of "0". The propensity score, i.e., the probability of being diagnosed with AdV was estimated via multivariable logistic regression, adjusting for the following baseline covariates: age, sex, birthweight, and number of people living in the same household. Continuous variables were modelled using restricted cubic splines, with three knots equally spaced. The estimated propensity score was then included as a covariate in the outcome model, which regressed the outcome on the type of virus of interest. This

propensity score approach was repeated for each type of virus [29]. Statistical analysis was performed using R version 4.2.0 software.

### Ethics statement and inclusivity in global research

This project was approved by the Sierra Leone Ethics and Scientific Review Committee (SLESRC April 3, 2020) and the Institutional Review Board at Vanderbilt University Medical Center (IRB# 201203). Written informed consent was obtained from the parent or legal guardian of all children enrolled in the study.

Additional information regarding the ethical, cultural, and scientific considerations specific to inclusivity in global research is included in the Supporting Information (S1 Checklist).

## Results

### Study population

From October 01, 2020 to October 31, 2021 a total of 1,226 children were screened with 695 confirmed eligible per criteria. Of these, 166 were not enrolled due to parent/guardian refusal. A total of 502 children less than 2-years of age were enrolled and included in this analysis.

### Demographic and clinical characteristics of hospitalized children

Of the 502 children enrolled, 239 (47.6%) were female with a median age of 8.6 months [IQR: 4.8; 14.2] (Table 1). Maternal education was low, with only 53.2% having achieved greater than a 6th grade level (primary education or less). Median household size was 8.0 persons [IQR: 5.0; 12.0] and the median number of children residing in each household that were less than two years of age was 2.0 [IQR 1.0; 3.0]. About 38.6% of parents reported that another household member also had an ARI in the two weeks prior to their child's admission to hospital. 39.6% of households reported having a smoker living in the household and only 32 households (6.4%) reported using an in-door wood burning stove.

Of the enrolled children, more than half were born premature (<37 weeks gestational age) (344, 68.5%) and the median birth weight of the enrolled children was 3.0 kilograms [IQR: 2.8; 3.4]. Seventy-three (14.5%) children enrolled were born by cesarean section. At the time of enrollment, 385 (76.7%) children reported that their diet still consisted of breastfeeding, of which 91 (23.6%) were exclusively breastfed, 209 (54.3%) were supplementing with formula feeds, and 85 (22.1%) were supplementing with solid foods. The median duration of breastfeeding was 8.0 months [IQR: 5.0; 13.0]. Malnutrition was common, with 186 children (37.3%) classified as having global acute malnutrition (weight for height z-score > -2) and 197 children (39.4%) classified as having global chronic malnutrition (height for age z-score > -2).

Human immunodeficiency virus (HIV), tuberculosis (TB), and sickle cell disease were uncommon in these children; however, 213 (42.4%) had a laboratory confirmed malaria diagnoses at the time of admission. Prior to hospitalization 35.3% of children had received antibiotics for this illness and 100% of children were administered antibiotics during their hospital stay. The median length of hospital stay was 3.0 days [IQR: 2.0; 6.0] and almost all (91.4%) were alive at discharge from the hospital.

### Respiratory virus detection

Of our 502 enrolled children, 376 (74.9%) had at least one respiratory virus detected by nasal swab testing with PCR. The most common virus isolated was HRV/EV (28.2%), followed by RSV (19.5%) and PIV (13.1%). Influenza and SARS-CoV-2 were identified in only 9.2% and 3.9% of children, respectively. Viral co-detection was common, with 97/376 (25.7%) having

**Table 1. Demographic and clinical characteristics of children under two-years hospitalized with respiratory symptoms, Kenema, October 2020-October 2021.**

| Variables (N = 502) | N (%) |
|---|---|
| **Sociodemographic characteristics** | |
| Sex | |
| Female | 239 (47.6%) |
| Age in months (median [IQR]) | 8.6 [4.8;14.2] |
| Mother's education | |
| No education | 166 (33.1%) |
| Primary education | 69 (13.7%) |
| Secondary education | 218 (43.4%) |
| Some or completed college | 49 (9.8%) |
| Number of household members, (median [IQR]) | 8.0 [5.0;12.0] |
| Number in household members aged < 5 years (median [IQR]) | 2.0 [1.0;3.0] |
| Other household member with acute respiratory infection in 2 weeks prior to admission | 194 (38.6%) |
| Smoker lives in the household | 199 (39.6%) |
| Household uses an indoor burning stove | 32 (6.4%) |
| **Birth characteristics** | |
| Premature, <37 weeks | 344 (68.5%) |
| Cesarean section | 73 (14.5%) |
| Birth weight in kilograms, (median [IQR]) (n = 501) | 3.0 [2.8;3.4] |
| **Nutrition characteristics** | |
| Currently breastfeeding | 385 (76.7%) |
| Exclusive breastfeeding | 91 (23.6%) |
| Supplementing with formula | 209 (54.3%) |
| Supplementing with solid foods | 85 (22.1%) |
| Duration of breastfeeding in months, (median [IQR]) (n = 491) | 8.0 [5.0;13.0] |
| Acute Malnutrition* (n = 498) | |
| Moderate (MAM) | 88 (17.7%) |
| Severe (SAM) | 98 (19.7%) |
| Global (GAM) | 186 (37.3%) |
| Chronic Malnutrition** (n = 500) | |
| Moderate (MCM) | 84 (16.8%) |
| Severe (SCM) | 113 (22.6%) |
| Global (GCM)** | 197 (39.4%) |
| **Diagnosis at admission** | |
| No respiratory virus detected (n = 502) | 126 (25.1%) |
| Respiratory virus detected[&] | 376 (74.9%) |
| ≥1 Respiratory viruses detected (n = 376) | 97 (25.7%) |
| Adenovirus (AdV) | 37 (7.4%) |
| Human Bocavirus (hBoV) | 26 (5.2%) |
| Endemic Human Coronaviruses (hCoV) | 28 (5.6%) |
| Human metapneumovirus (hMPV) | 24 (4.8%) |
| Influenza (A, B, C) | 46 (9.2%) |
| Parainfluenza viruses (PIV 1–4) | 66 (13.1%) |
| Human Rhinovirus/Enterovirus (HRV/EV) | 142 (28.2%) |
| Respiratory Syncytial Virus (RSV) | 98 (19.5%) |
| SARS-CoV-2 | 20 (3.9%) |
| Other diagnosis and/or treatment | |

*(Continued)*

**Table 1.** (Continued)

| Variables (N = 502) | N (%) |
|---|---|
| Malaria (positive rapid diagnostic test or blood smear result) | 213 (42.4%) |
| Antimalarials given | 292 (58.2%) |
| Malaria + viral co-infection (n = 213) | 147 (69.0%) |
| HIV/AIDS test performed | 66 (13.1%) |
| HIV/AIDS: positive | 2 (3.0%) |
| Tuberculosis | 7 (13.1%) |
| Sickle cell test performed | 36 (7.2%) |
| Sickle cell: positive | 7 (19.4%) |
| Antibiotics given prior to hospitalization with this illness | 177 (35.3%) |
| Antibiotics given during hospitalization | 502 (100.0%) |
| **Respiratory symptom severity** | |
| Mild | 403 (80.2%) |
| Moderate | 86 (17.1%) |
| Severe | 13 (2.5%) |
| Received Supplemental Oxygen | 170 (33.8%) |
| Referred to intensive care unit (ICU) | 303 (60.3%) |
| **Status at discharge** | |
| Alive | 459 (91.4%) |
| Dead | 24 (4.8%) |
| Discharged against medical advice/abscond | 19 (3.8%) |
| Length of stay in hospital in days (**median [IQR]**) | 3.0 [2.0: 6.0] |

*Moderate acute malnutrition = weight for height z-score between -2 and -3

*Severe acute malnutrition = weight for height z-score > -3

*Global acute malnutrition = weight for height z-score > -2 (moderate + severe)

**Moderate Chronic malnutrition = height for age z-score between -2 and -3

**Severe Chronic malnutrition = height for age z-score > -3

**Global Chronic malnutrition = height for age z-score > -2 (moderate + severe)

&Of 376 participants with a viral infection detected, a total of 487 viruses were isolated. Percentages of virus' represents the proportion of a given virus out of the 502 total number of participants.

more than one virus identified (Fig 1). Of the 213 children with a laboratory confirmed malaria diagnosis, 147 (69.5%) had a virus identified as well.

The demographic and clinical characteristics of enrolled children were compared based on viral detection by PCR testing (Table 2). Children with a virus detected were younger (median age 8.13 months [IQR: 4.47; 13.70] vs. 10.5 months [IQR: 6.15; 15.60]; p = 0.009); had a shorter duration of reported breastfeeding (median 7.0 months [IQR 4.0; 12.0] vs. 9.5 months [IQR: 5.0; 14.0], p = 0.025); were less frequently classified as having global chronic malnutrition (36.5% vs. 48.0%, p = 0.030); and had a lower frequency of malaria co-infection at the time of admission (39.4% vs. 51.6%, p = 0.022).

## Seasonality of respiratory viruses

HRV/EV, AdV, and PIV all had a fairly steady year-round circulation with only a slight decline in cases seen between August to October. All other viruses showed more distinct seasonal peaks (Fig 2). For example, the endemic common cold hCoV's and hBoV showed peaks in cases between January and April, during a time that corresponds with Sierra Leone's typical dry season (December to April). In contrast, influenza, RSV, and hMPV showed a peak in

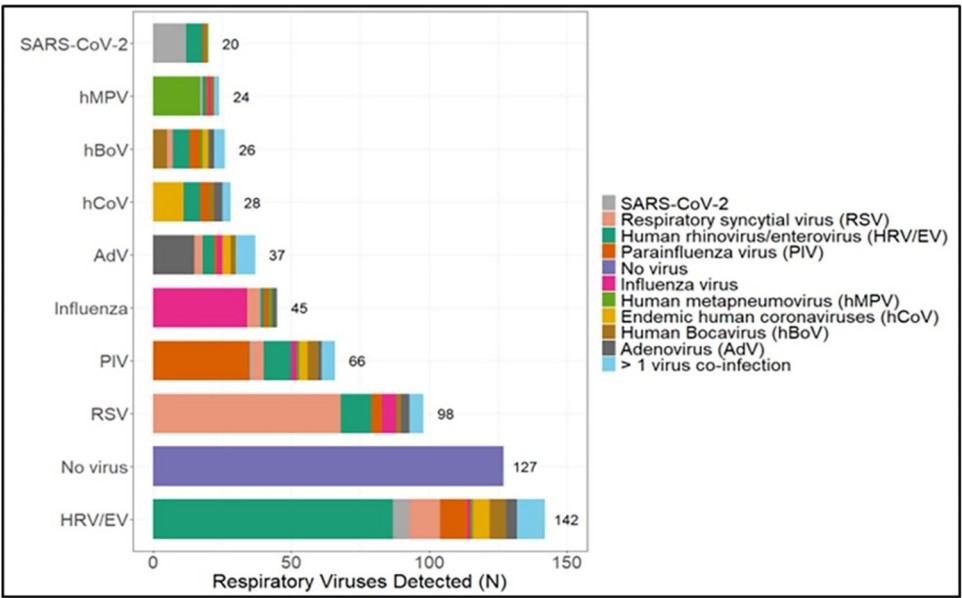

**Fig 1. Total number of viral pathogens identified and their associated co-detected viruses.**

cases that corresponded more to Sierra Leone's wet season (May to November), with hMPV appearing earlier in June and July, while RSV and influenza peaked later from August to November.

Overall, the number of cases of SARS-CoV-2 in our population was low (only 20 total), despite the fact that enrollment into our study corresponded to a period of intense pandemic escalation. In our patients, SARS-CoV-2 detection was predominantly limited to January 2021 and a second peak in June and July 2021.

## Determinants of need for O2 therapy, referral to the ICU, and increased severity score

In multivariable analysis, we explored patient characteristics associated with the need for O2 therapy administration, the need for referral to a higher level of care in the ICU, and their impact on severity score (Table 3). Children who were of younger age (6 months vs. 12 months of age) had a roughly two-fold higher likelihood of requiring O2 therapy (aOR 2.02, 95% CI:1.57, 2.58, p = 0.007) than older children. Similar results were observed for referral to the ICU and severity scores, although the overall effect of age in both regressions was not statistically significant (p>0.05); younger patients (6 months compared to 12 months) were more likely to require ICU (aOR 1.64, 95%CI:1.24, 2.18) and to have higher severity score (aOR 1.86, 95%CI: 1.51, 2.28). Further, children with more than one virus detected had an approximately 1.7-fold higher odds of requiring O2 therapy during their hospitalization compared to those who had a single virus detected. For children with malnutrition, we saw mixed results. Children classified as having Global Acute Malnutrition had a 61% lower odds of being transferred to the ICU (aOR 0.39, 95%CI: 0.26, 0.61, p<0.001), while those classified as having Global Chronic Malnutrition had 53% lower odds of being transferred to the ICU (aOR 0.47, 95%CI: 0.30, 0.72, p = 0.001). Children that received antibiotics prior to admission had also higher odds of having higher severity scores (aOR 1.41, 95%CI: 1.01, 1.98, p = 0.042). We also performed sensitivity analysis to assess the effect of multicollinearity between acute and chronic malnutrition. We ran additional regression models, adjusting for each variable at a

**Table 2. Demographic and clinical characteristics of children under two-years, by whether a virus was detected on PCR, hospitalized with respiratory symptoms, Kenema, October 2020-October 2021.**

| Variables (N = 502) | Virus-positive 376 (75%) | Virus-negative 126 (25%) | p-value |
|---|---|---|---|
| **Sociodemographic characteristics** | | | |
| Sex | | | 0.605 |
| Female | 176 (46.8%) | 63 (50.0%) | |
| Age in months (**median [IQR]**) | 8.13 [4.47;13.7] | 10.5 [6.15;15.6] | **0.009** |
| Mother's education | | | 0.338 |
| No education | 118 (31.4%) | 48 (38.1%) | |
| Primary education | 52 (13.8%) | 17 (13.5%) | |
| Secondary education | 165 (43.9%) | 53 (42.1%) | |
| Some or completed college | 41 (10.9%) | 8 (6.4%) | |
| Number of household members, (**median [IQR]**) | 8.0 [5.0;12.0] | 8.0 [5.0;11.] | 0.724 |
| Number of household members aged < 5 years (**median [IQR]**) | 2.0 [1.0;3.0] | 2.00 [1.0;2.0] | 0.112 |
| Other household member with acute respiratory infection in 2-weeks prior to admission | 146 (38.8%) | 48 (38.1%) | 0.967 |
| Smoker lives in the household | 149 (39.6%) | 50 (39.7%) | 0.493 |
| Household uses an indoor burning stove | 24 (6.4%) | 8 (6.4%) | 1.000 |
| **Birth characteristics** | | | |
| Premature, <37 weeks | 262 (69.7%) | 82 (65.1%) | 0.394 |
| Cesarean section | 51 (13.6%) | 22 (17.5%) | 0.354 |
| Birth weight in kilograms, (**median [IQR]**) (**n = 501**) | 3.0 [2.8;3.4] | 3.0 [2.7;3.3] | 0.344 |
| **Nutrition characteristics** | | | |
| Duration of breastfeeding in months, (**median [IQR]**) (**n = 491**) | 7.0 [4.0;12.0] | 9.5 [5.0;14.0] | **0.025** |
| Global Acute Malnutrition (GAM)* (**n = 498**) | 135 (36.2%) | 51 (41.1%) | 0.381 |
| Global Chronic Malnutrition (GCM)** (**n = 500**) | 137 (36.5%) | 60 (48.0%) | 0.030 |
| **Other diagnosis and/or treatment** | | | |
| Malaria (positive rapid diagnostic test or blood smear result) | 148 (39.4%) | 65 (51.6%) | **0.022** |
| Antimalarials given | 212 (56.4%) | 80 (63.5%) | 0.195 |
| Antibiotics given prior to hospitalization with this illness | 140 (37.2%) | 37 (29.4%) | 0.136 |
| Antibiotics given during hospitalization | 376 (100%) | 126 (100%) | 1.001 |
| **Status at discharge** | | | 0.492 |
| Alive | 344 (91.5%) | 115 (91.3%) | |
| Dead | 17 (4.5%) | 7 (5.6%) | |
| Discharged against medical advice/abscond | 15 (3.9%) | 4 (3.2%) | |
| Length of Stay in Hospital in Days | 4.0 [2.0;6.0] | 3.0 [2.0;5.0] | 0.388 |

*Global Acute Malnutrition = weight for height z-score > -2

**Global Chronic Malnutrition = height for age z-score > -2

time, and assessed its effect on the outcomes of interest. The results were the same as when both variables were included simultaneously in the models and are thus omitted from presentation.

When we explored associations between virus type and need for O2 therapy, ICU referral, or increased severity scores, we found both hMPV (aOR 2.61, 95%CI:1.09, 6.21, p = 0.029) and RSV (aOR 2.03, 95%CI:1.26, 3.27, p = 0.003) had more than two-fold higher odds of requiring O2 therapy while hospitalized (Table 4). No associations were found between virus type and referral to the ICU. Influenza was associated with lower severity scores (aOR 0.58, 95%CI: 0.35, 0.99 p = 0.045). Although, on average, patients with RSV had higher odds of increased severity scores (OR 1.39, 95%CI: 0.93, 2.07, p = 0.106), this association was not statistically significant (p>0.05).

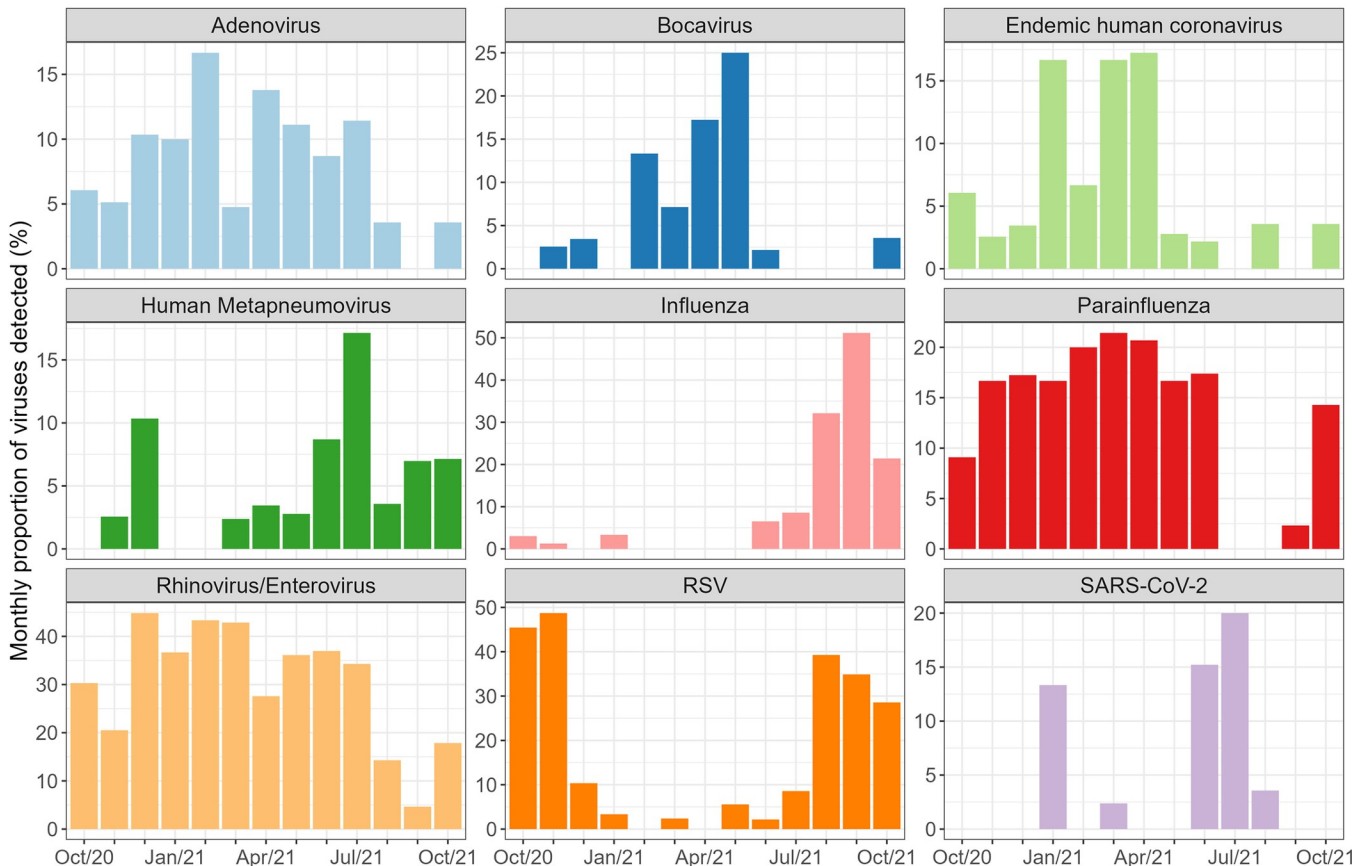

**Fig 2. Respiratory virus detection by month (October 2020 to October 2021).**

**Table 3. Multivariable logistic regression for variables associated with need for oxygen, referral to the ICU, and increasing severity score.**

| | O2 required | | Referral to ICU | | Severity Score | |
|---|---|---|---|---|---|---|
| **Variables** | **aOR (95% CI)** | **p-value** | **aOR (95% CI)** | **p-value** | **aOR (95% CI** | **p-value** |
| **Age** | | **0.007** | | **0.187** | | **0.064** |
| **6 months** | **2.02 (1.57, 2.58)** | | **1.64 (1.24, 2.18)** | | **1.86 (1.51, 2.28)** | |
| 12 months | Ref | | Ref | | Ref | |
| 18 months | 0.95 (0.64,1.41) | | 0.85 (0.59, 1.22) | | 0.77 (0.57, 1.03) | |
| Male | 0.71 (0.47, 1.08) | 0.110 | 1.13 (0.75, 1.72) | 0.558 | 0.91 (0.19, 4.35) | 0.553 |
| Prematurity (<37 weeks) | 1.33 (0.85, 2.08) | 0.206 | 1.30 (0.84, 2.02) | 0.247 | 1.21 (0.72, 2.03) | 0.287 |
| Virus co-infection (more than one virus) | 1.71 (1.04, 2.80) | 0.033 | 0.69 (0.41, 1.14) | 0.145 | 1.18 (0.53, 2.63) | 0.419 |
| Exposure to indoor pollutants* | 1.17 (0.78, 1.75) | 0.462 | 0.81 (0.54, 1.22) | 0.308 | 1.24 (0.90, 1.72) | 0.186 |
| Global Acute Malnutrition[&] | 1.27 (0.83, 1.94) | 0.276 | 0.39 (0.26, 0.61) | <0.001 | 1.22 (0.88, 1.68) | 0.251 |
| Global Chronic Malnutrition[@] | 0.82 (0.52, 1.27) | 0.370 | 0.47 (0.30, 0.72) | 0.001 | 0.79 (0.56, 1.13 | 0.187 |
| Length of stay in Hospital in days | 1.04 (0.99, 1.08) | 0.104 | 0.99 (0.96, 1.02) | 0.496 | 1.01 (0.67, 1.52) | 0.331 |
| Antibiotics received prior to admission | 1.16 (0.76, 1.77) | 0.478 | 0.74 (0.48, 1.13) | 0.160 | 1.41 (1.01, 1.98) | 0.042 |

Ref: Reference level

[&] Global Acute Malnutrition = weight for height z-score > -2

[@] Global Chronic Malnutrition = height for age z-score > -2

*Exposure to indoor pollutants: People living with a smoker and use of an indoor burning stove

**Table 4. Multiple logistic regression with propensity score adjustment for the association between virus and worsening clinical condition.**

| Viruses | O2 required | | Referred to ICU | | Severity Score | |
|---|---|---|---|---|---|---|
| | aOR (95% CI) | p-value | aOR (95% CI) | p-value | aOR (95% CI) | p-value |
| AdV | 1.20 (0.58, 2.47) | 0.615 | 0.56 (0.28, 1.12) | 0.100 | 0.93 (0.51, 1.68) | 0.799 |
| hBoV | 0.80 (0.34, 1.90) | 0.620 | 0.50 (0.22, 1.12) | 0.090 | 1.03 (0.51, 2.10) | 0.932 |
| hCoV | 1.04 (0.46, 2.37) | 0.916 | 1.21 (0.54, 2.71) | 0.633 | 1.22 (0.62, 2.40) | 0.569 |
| SARS-CoV-2 | 2.10 (0.85, 5.02) | 0.108 | 0.65 (0.26, 1.61) | 0.360 | 1.85 (0.79, 4.33) | 0.157 |
| Influenza A, B, and C | 0.69 (0.34, 1.42) | 0.318 | 1.26 (0.66, 2.43) | 0.477 | 0.58 (0.35, 0.99) | 0.045 |
| hMPV | 2.61 (1.09, 6.21) | 0.029 | 2.11 (0.75, 5.87) | 0.151 | 1.18 (0.57, 2.43) | 0.661 |
| PIV 1–4 | 1.07 (0.61, 1.87) | 0.803 | 0.89 (0.52, 1.52) | 0.684 | 1.33 (0.83, 2.13) | 0.235 |
| HRV/EV | 0.87 (0.58,1.32) | 0.519 | 0.98 (0.65, 1.46) | 0.936 | 0.97 (0.69, 1.37) | 0.882 |
| RSV | 2.03 (1.26, 3.27) | 0.003 | 0.81 (0.50, 1.32) | 0.410 | 1.39 (0.93, 2.07) | 0.106 |

AdV = adenovirus; hBoV = Human Bocavirus; hCoV = Human endemic coronaviruses; hMPV = Human metapneumovirus; PIV = parainfluenza virus; HRV/
EV = Human rhinovirus/enterovirus; RSV = respiratory syncytial virus.

Propensity score model adjusted for age, gender, birthweight, and number of persons living in the household.

## Discussion

Repeated global outbreaks of respiratory virus infections have had negative impacts on overall population-level health as well as placing significant strains on the global economy and national development efforts [30]. The recent pandemic of severe acute respiratory syndrome coronavirus 2 (SARS-CoV-2) and associated novel coronavirus disease 2019 (COVID-19) highlight this point and underscore the urgency for strengthening global public health surveillance for early detection and control of respiratory viruses.

Our study describes the prevalence and temporality of viral respiratory pathogens identified in children less than 24 months of age who were admitted to hospital between October 2020 and October 2021 in rural eastern Sierra Leone with acute respiratory symptoms. This period corresponds to the latter part of the first year of the COVID-19 pandemic and much of the pandemic's second year. Viral infection incidence and temporality are impacted by a variety of factors including precipitation; temperature and humidity; hemisphere location; as well as human behavior and the impact of changing environmental conditions on host defense mechanisms [31, 32]. To date, quite a bit is known about the epidemiology and seasonal variation of viruses in temperate climates. For example, influenza, RSV and the endemic hCoV's are well described to typically occur in the colder, dryer winter months in temperate climates, while viruses such as PIV, hMPV, and HRV/EV are more likely to peak during the spring and early summer months [32–34]. In contrast, a much smaller number of studies have reported on the seasonal variation of respiratory viruses in tropical countries [7, 32, 35]. As such, one of the goals of our study was to provide baseline data on respiratory virus prevalence among a younger pediatric population, in a tropical region of West Africa, in order to aid health authorities with decision making around surveillance and prevention.

In our study, nearly three-quarters of the children had at least one virus detected. This is comparable to many other surveillance studies both in developed and developing countries. The overall prevalence of these viruses is also comparable to previous studies which have highlighted RSV, PIV, influenza, and hMPV as the most common viral etiologies of ARI among children in the under-five year age group [33, 36, 37].

HRV/EV and PIV's were two of the top three viruses detected in our study population. Together with AdV, which was detected in 7.4% of children, and the endemic hCoV's, which were detected in 5.6%, these viruses circulated mostly year-round, without distinct seasonal

variability. Each of these viruses, however, showed a distinct decline in cases corresponding with the main peak in influenza cases in our study (August to October 2021). Respiratory virus interference, in which respiratory virus transmission can interfere or inhibit the transmission of other respiratory viruses, has been previously described, especially in relation to peaks in influenza cases and may have contributed to the pattern of seasonal variability seen in our study population [38–40].

Multiple reports have described that infection with a respiratory virus generally follows a recognizable seasonal pattern in temperate climates, whereas they may circulate all year round in tropical regions [41–44]. In these studies, peaks in RSV and influenza cases have tended to correspond to winter months in temperate regions, while showing greater diversity in temporality in tropical regions. Furthermore, peaks in RSV and influenza cases have occurred later in the year, comparatively, as geographic latitude increases. In tropical countries that have distinct wet and dry seasons, the peak in cases of RSV and influenza have tended to correspond mostly with the wet, rainy season [7, 36, 45]. In Sierra Leone, the wet season occurs each year during the period from May to November, with the heaviest rain fall typically between July and September. This is then followed by a dry season from December to April [46]. During our period of study, this predilection for cases peaking in the wet season was similarly observed, with RSV cases peaking between October and November in 2020. Then in 2021, both RSV and influenza cases peaked during the heaviest rain fall months between August and October. It has been well documented that there were reductions in the annual epidemics of most seasonal respiratory viruses in 2020/2021, likely as a result of the COVID-19 pandemic and the large scale strategies implemented worldwide that year in an attempt to mitigate COVID-19 [13, 47, 48]. Influenza in particular had a notable absence globally during the 2020/2021 seasons, and in the southern hemisphere in particular, showed very low levels of activity from May to September 2020 during the southern hemisphere´s typical winter months [12, 49]. While our study period only catches the last few months of Sierra Leone´s typical 2020 influenza period, the absence of influenza cases in 2020 appears to have been similar in our study population.

Children enrolled in our study were younger (median age 8.6 months) and had a high rate of respiratory virus co-infection (19.3% of all children enrolled and 25.7% among those diagnosed with any respiratory virus infection). In previous studies, younger children, especially those less than one year of age, similarly showed both higher rates of viral ARI overall as well as viral co-infection [6, 50–53]. This is likely due to a developing immune system and poor ability for viral elimination [54–56]. While data on respiratory virus co-infection remains limited, reports have described that such infections may be more severe compared to single pathogen infections [57, 58]. This was the case in our study as well, where in multivariable analysis, children with more than one virus detected had a higher odds of requiring oxygen therapy during their hospitalization (aOR 1.71, p = 0.033). Furthermore, the likelihood of needing supplemental oxygen therapy was two-fold higher in children who were younger (less than a year old) compared with older children (aOR 2.02, p = 0.007) and for children diagnosed with RSV (aOR 2.03, p = 0.003) and hMPV (aOR 2.61, p = 0.029).

Maternal education of our children was low. Studies done in comparable resource limited settings have shown similar trends in ARI frequency and disease severity in children whose mothers had lower educational attainment, possibly due to more constant physical contact between mother and child and due to the fact that knowledge of, and access to, preventive and/or therapeutic measures may be lacking [59–61]. Although this study did not show correlations between certain well-known risk factors (such as overcrowding, living with smokers in the same household, presence of indoor wood burning stove), these risk factors were documented in children presenting to the hospital with respiratory symptoms. These risk factors,

although typical to these low-income settings, are modifiable, and interventions such as targeted vaccinations of vulnerable groups and continuous community engagement to improve vaccine uptake of these groups and social lifestyle could help mitigate the increased presentations of respiratory symptoms to the hospitals.

Lack of breastfeeding has been shown to put children under-five years of age at increased risk of ARI severity and hospitalizations due to the child not receiving boosted immunity from the mother [50, 62]. In our study, children with a virus detected at enrollment were more likely to have breastfed for less time (median of 7 months vs. 9.5 months; p = 0.03). In settings such as Sierra Leone, breastfeeding is already commonly practiced and significant investments have been made in the promotion of exclusive breastfeeding until six-months of age [63, 64]. Perhaps these strategies should be re-evaluated, promoting breastfeeding for longer periods of time.

Malnutrition was common in our study population, with about 37.3% and 39.4% of children categorized as having either global acute malnutrition or global chronic malnutrition respectively. Among children categorized as malnourished, more than a third were identified as having a viral infection at enrollment. Further, those categorized as having global chronic malnutrition had a 2-fold higher likelihood of being referred to the ICU during this hospitalization (aOR 2.14, p = 0.001). While we cannot determine in our study if the referral to ICU was for nutrition related issues or due to respiratory compromise or both, there is strong evidence in the literature to support the association between inadequate growth and the risk of severe disease and/or death from an acute lower respiratory infection in early childhood [65–68].

One of the key strengths of our study was having a rich data set to describe respiratory viral surveillance, in a previously unstudied pediatric population, during much of the early part of the COVID-19 pandemic. Prior to this study, no large prospective studies of respiratory viral disease had been completed in this setting. Our study, however, has some limitations as well. The KGH study site, despite being the main referral facility for sick children in Eastern Province, Sierra Leone does not receive all pediatric referrals. There are other centers with high pediatrics admissions. Furthermore, only hospitalized children were enrolled into the study, and in addition, enrollment was conducted through convenience sampling. As a result, we are unable to categorize respiratory virus prevalence among children seen in the outpatient clinics, and therefore, our results may not be a true reflection of the respiratory viral prevalence in the province, nor generalizable to the country as a whole. Our ongoing surveillance allowed us to collect samples over the course of a year, in order to begin describing the seasonality of respiratory virus prevalence in hospitalized children in this region. However, while the timing of our study allowed us a unique opportunity to describe the prevalence of respiratory viruses in young children during the global COVID-19 pandemic, we acknowledge that this also likely has had a dramatic impact on the generalizability of our results for estimating across future years. Moreover, a realistic description of the seasonal variation in respiratory virus cases more broadly, is more complicated than can be assessed in just one year. Lastly, bacterial culture capacity is non-existent in Kenema such that we were unable to analyze for bacterial causes of ARI in our patients.

## Conclusion

Viral pathogen prevalence was high (74.9%) in children under-two years of age, hospitalized with respiratory symptoms in Eastern Province, Sierra Leone. Viral co-detection with more than one virus and viral co-detection with malaria were also common. Influenza, RSV and hMPV showed peaks in cases that corresponded with Sierra Leone´s typical wet season (May

to November), while the endemic common cold hCoV's and hBoV showed peaks in cases corresponding to Sierra Leone´s typical dry season (December to April). HRV/EV, AdV, and PIV each had a fairly steady year-round circulation. Despite a high prevalence of respiratory viruses isolated, 100% of children received antibiotics, underscoring a need to expand laboratory diagnostic capacity and to revisit clinical guidelines and their implementation in the management of children presenting to low-resource facilities with respiratory symptoms. SARS-CoV-2 identification in our patients was quite low, and this finding warrants further exploration. However, the frequency of other viruses reported in our study, especially influenza, were consistent with what was reported elsewhere and may represent alterations in their seasonality as a result of the COVID-19 pandemic, though further study for longer periods of time is needed to fully elucidate these trends. Continuous surveillance and serologic studies among more diverse age groups, with greater geographic breadth, are needed in Sierra Leone to better characterize the long-term impact of COVID-19 on respiratory virus prevalence and to better characterize the seasonality of respiratory virus transmission in general in Sierra Leone.

## Supporting information

**S1 Checklist. This file is the inclusivity in global research checklist.**
(DOCX)

## Acknowledgments

The authors are very grateful to the study subjects for their participation in this study, as well as the staff and management of Kenema Government Hospital.

## Author Contributions

**Conceptualization:** Robert J. Samuels, Zaid Haddadin, Natasha B. Halasa.

**Data curation:** Ibrahim Sumah, Foday Alhasan.

**Formal analysis:** Robert J. Samuels, Gustavo Amorim, John S. Schieffelin, Troy D. Moon.

**Investigation:** Ibrahim Sumah, Rendie McHenry, Laura Short, James D. Chappell, Natasha B. Halasa.

**Methodology:** Zaid Haddadin, Natasha B. Halasa, Troy D. Moon.

**Project administration:** Ibrahim Sumah.

**Resources:** Natasha B. Halasa, John S. Schieffelin, Troy D. Moon.

**Supervision:** Donald S. Grant, John S. Schieffelin, Troy D. Moon.

**Validation:** Gustavo Amorim.

**Writing – original draft:** Robert J. Samuels.

**Writing – review & editing:** Zaid Haddadin, Natasha B. Halasa, Inaê D. Valério, Gustavo Amorim, John S. Schieffelin, Troy D. Moon.

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
