## [Decision Letter · Decision Letter 0]

30 May 2023

PONE-D-23-09189Respiratory virus surveillance in hospitalized children less than two-years of age in Kenema, Sierra Leone during the COVID-19 pandemic (October 2020- October 2021)PLOS ONE

Dear Dr. Samuels,

Thank you for submitting your manuscript to PLOS ONE. After careful consideration, we feel that it has merit but does not fully meet PLOS ONE’s publication criteria as it currently stands. Therefore, we invite you to submit a revised version of the manuscript that addresses the points raised during the review process.

The Authors are expected to address all the criticisms by the Reviewer. In particular, please clarify the age categories in the analysis and provide more clarification and discussion for the analysis on the variables “Global Acute Malnutrition” and “Global Chronic Malnutrition”.

We look forward to receiving your revised manuscript.

Kind regards,

Eric HY Lau, Ph.D.

Academic Editor

PLOS ONE

Additional Editor Comments:

The Authors are expected to address all the criticisms by the Reviewer. In particular, please clarify the age categories in the analysis and provide more clarification and discussion for the analysis on the variables “Global Acute Malnutrition” and “Global Chronic Malnutrition”.

Reviewers' comments:

Reviewer's Responses to Questions

**Comments to the Author**

1. Is the manuscript technically sound, and do the data support the conclusions?

Reviewer #1: Yes

2. Has the statistical analysis been performed appropriately and rigorously? 

Reviewer #1: Yes

3. Have the authors made all data underlying the findings in their manuscript fully available?

Reviewer #1: Yes

4. Is the manuscript presented in an intelligible fashion and written in standard English?

Reviewer #1: Yes

5. Review Comments to the Author

Reviewer #1: This is a very interesting manuscript where the authors describe the prevalence and temporality of viral respiratory pathogens in Sierra Leone during the second year of the COVID-19 pandemic. The manuscript is very well written and utilizes sound methods which are well described.

Here are some comments for the authors to consider:

1. The authors, throughout the text, do report data to one decimal place in some cases and not so in other instances. For example, in line221 they write 36.5% vs 48%. Consider reviewing and reporting all data to one decimal place throughout the text and tables.

2. In Fig 3, can they consider presenting these as the percent of tests conducted each month instead of frequencies?

3. In lines 246-247, the authors report that children who “were of younger age (6 months vs. 12 months of age) had a roughly two-fold higher likelihood of requiring oxygen therapy……“. These data are also presented in Table 3. It is not clear to me how the data were analyzed, did the authors mean to use age categories of 0-5, 6-11, 12-17 and 18-23 months? Please clarify or review the analyses as appropriate.

4. In Table 3, the authors present data for “Exposure to indoor pollutants”. I guess they considered living with a smoker and use of an indoor burning stove. Can they provide a footnote that defines what this variable is? The authors provide a footnote “*Adjusted for age, gender……” but the symbol (*) is not indicated anywhere on the table itself nor title.

5. In lines 253 – 256, the authors report that they “saw mixed results” for children classified as having “Global Acute Malnutrition” which seemed to have a protective effect compared to those with “Global Chronic Malnutrition” which seemed to be a risk factor for ICU admission. Can the authors consider discussing these results? It doesn’t appear as though both variables were included in the multivariable model as the authors indicate that they only adjusted for age, gender, birthweight, and number of persons living in the household. However, if it so happened that both variables were included in the multivariable model then the authors might want to investigate a possible effect of multicollinearity.

6. In Table 4, the authors provide a footnote “*Propensity score model…..” but the symbol (*) is not indicated anywhere on the table itself nor title.

7. In lines 404-406, the authors state “However, likely alterations in the seasonality of the other respiratory viruses, especially influenza, were consistent with what was reported elsewhere and felt to be a result of the COVID-19 pandemic”. With only one years’ worth of data and without presenting any data prior to the COVID-19 pandemic, I don’t think authors can infer alterations in seasonality. Consider rephrasing and also providing references to the data “reported elsewhere”.

6. PLOS authors have the option to publish the peer review history of their article (what does this mean?). If published, this will include your full peer review and any attached files.

Reviewer #1: No

---

## [Author Response · Author response to Decision Letter 0]

2 Sep 2023

# Review 1: 

This is a very interesting manuscript where the authors describe the prevalence and temporality of viral respiratory pathogens in Sierra Leone during the second year of the COVID-19 pandemic. The manuscript is very well written and utilizes sound methods which are well described.

R: We appreciate the reviewer´s comments

The authors, throughout the text, do report data to one decimal place in some cases and not so in other instances. For example, in line221 they write 36.5% vs 48%. Consider reviewing and reporting all data to one decimal place throughout the text and tables.

R: Thank you. We agree with the reviewer and have updated the tables.

In Fig 3, can they consider presenting these as the percent of tests conducted each month instead of frequencies?

R: The authors have updated Figure 3 per the reviewer´s recommendations

In lines 246-247, the authors report that children who “were of younger age (6 months vs. 12 months of age) had a roughly two-fold higher likelihood of requiring oxygen therapy……“. These data are also presented in Table 3. It is not clear to me how the data were analyzed, did the authors mean to use age categories of 0-5, 6-11, 12-17 and 18-23 months? Please clarify or review the analyses as appropriate.

R: We thank the reviewer for this comment and the opportunity to clarify the comparison. We did not disaggregate age into age groups; instead, we analyzed age as a continuous variable and modelled age using restricted cubic splines to alleviate linearity assumptions. We have added text to the Methods section, lines 152-157 of the track changes version to better clarify.

In Table 3, the authors present data for “Exposure to indoor pollutants”. I guess they considered living with a smoker and use of an indoor burning stove. Can they provide a footnote that defines what this variable is? The authors provide a footnote “*Adjusted for age, gender……” but the symbol (*) is not indicated anywhere on the table itself nor title.

R: Thank you for noting this. Per the reviewer´s recommendations we have added a footnote to Table 3(See line 280 in the track changes version). Thank you also for bringing our attention to the symbol (*). It should not be there, and this sentence was removed (line 277 removed in the track changes version).

In lines 253 – 256, the authors report that they “saw mixed results” for children classified as having “Global Acute Malnutrition” which seemed to have a protective effect compared to those with “Global Chronic Malnutrition” which seemed to be a risk factor for ICU admission. Can the authors consider discussing these results? It doesn’t appear as though both variables were included in the multivariable model as the authors indicate that they only adjusted for age, gender, birthweight, and number of persons living in the household. However, if it so happened that both variables were included in the multivariable model then the authors might want to investigate a possible effect of multicollinearity.

R: We thank the reviewer for bringing our attention to this point and having a chance to review. Carefully looking at the results, we noticed that the reference level for chronic malnutrition was reversed, so that the odds ratio reported was also reversed: Instead of seeing an odds ratio of 2.14, the actual value with ‘normal’ as the reference group is 0.47 (=1/2.14). This also affects the other regressions (O2 required and Severity Score). This has been corrected in Table 3 and in the discussions following that (See line 260-262 in the track changes version). We apologize for the mistake and thank the reviewer for his critical view of our findings.

To clarify the issue of multicollinearity, we added additional text to the Results Section lines 265 – 269 in the track change version. 

In Table 4, the authors provide a footnote “*Propensity score model…..” but the symbol (*) is not indicated anywhere on the table itself nor title.

R: Thank you. We have removed the (*) symbol from the footnotes. .

In lines 404-406, the authors state “However, likely alterations in the seasonality of the other respiratory viruses, especially influenza, were consistent with what was reported elsewhere and felt to be a result of the COVID-19 pandemic”. With only one years’ worth of data and without presenting any data prior to the COVID-19 pandemic, I don’t think authors can infer alterations in seasonality. Consider rephrasing and also providing references to the data “reported elsewhere”.

R: Per the reviewer´s recommendations the text has been altered at lines 421 – 424 of the track changes versions to better clarify our intended meaning.

When submitting your revision, we need you to address these additional requirements. 1. Please ensure that your manuscript meets PLOS ONE's style requirements, including those for file naming. The PLOS ONE style templates can be found at.

R: Formatting has been updated

R: The questionnaire for Inclusivity in Global Research has been completed and uploaded.

R: a deidentified data set has been uploaded to https://osf.io/2peh6/files/osfstorage/6493215b67aff805b3ee0303

R: Figure 1 has been removed from the manuscript

---

## [Editor Report · Decision Letter 1]

27 Sep 2023

Respiratory virus surveillance in hospitalized children less than two-years of age in Kenema, Sierra Leone during the COVID-19 pandemic (October 2020- October 2021)

PONE-D-23-09189R1

Dear Dr. Samuels,

We’re pleased to inform you that your manuscript has been judged scientifically suitable for publication and will be formally accepted for publication once it meets all outstanding technical requirements.

Kind regards,

Eric HY Lau, Ph.D.

Academic Editor

PLOS ONE

Additional Editor Comments (optional):

Thanks for addressing all the reviewers' comments. Congratulations on the excellent work!
---

## [Editor Report · Acceptance letter]

2 Oct 2023

PONE-D-23-09189R1 

Respiratory virus surveillance in hospitalized children less than two-years of age in Kenema, Sierra Leone during the COVID-19 pandemic (October 2020- October 2021) 

Dear Dr. Samuels:

I'm pleased to inform you that your manuscript has been deemed suitable for publication in PLOS ONE. Congratulations! Your manuscript is now with our production department. 

Kind regards, 

on behalf of

Dr. Eric HY Lau 

Academic Editor

PLOS ONE